# Predictors of Humoral Response to mRNA COVID19 Vaccines in Kidney Transplant Recipients: A Longitudinal Study—The COViNEPH Project

**DOI:** 10.3390/vaccines9101165

**Published:** 2021-10-12

**Authors:** Alicja Dębska-Ślizień, Zuzanna Ślizień, Marta Muchlado, Alicja Kubanek, Magdalena Piotrowska, Małgorzata Dąbrowska, Agnieszka Tarasewicz, Andrzej Chamienia, Bogdan Biedunkiewicz, Marcin Renke, Leszek Tylicki

**Affiliations:** 1Department of Nephrology Transplantology and Internal Medicine, Faculty of Medicine, Medical University of Gdańsk, 80-210 Gdańsk, Poland; adeb@gumed.edu.pl (A.D.-Ś.); zuzanna.slizien@gumed.edu.pl (Z.Ś.); marta.muchlado@gumed.edu.pl (M.M.); ataras@gumed.edu.pl (A.T.); chamien@gumed.edu.pl (A.C.); bogdan.biedunkiewicz@gumed.edu.pl (B.B.); 2Department of Occupational, Metabolic and Internal Diseases, Faculty of Health Science, Medical University of Gdansk, 81-519 Gdynia, Poland; alicja.kubanek@gumed.edu.pl (A.K.); mrenke@gumed.edu.pl (M.R.); 3Department of Medical Immunology, Faculty of Medicine, Medical University of Gdańsk, 80-210 Gdańsk, Poland; m.piotrowska@gumed.edu.pl; 4Central Clinical Laboratory, The University Clinical Centre, 80-952 Gdańsk, Poland; mdabrowska@uck.gda.pl

**Keywords:** COVID-19, kidney transplant recipients, mRNA vaccines, seroconversion

## Abstract

Background: The efficacy of SARS-CoV-2 vaccination among kidney transplant recipients (KTR) is low. The main goal of this study was to analyze factors that may influence the humoral response to vaccination. Methods: We analyzed the titer magnitude of IgG antibodies directed against spike (S)-SARS-CoV-2 antigen after the second dose of the mRNA vaccine in 142 infection naïve KTR (83 men, i.e., 58.4%) with a median age (IQR) of 54 (41–63), and 36 respective controls without chronic kidney disease. mRNA-1273 or BNT162b2 were applied in 26% and 74% of KTR, respectively. Results: S-specific immune response (seroconversion) was seen in 73 (51.41%) of KTR, and in all controls 36 (100%). Independent predictors of no response were elder age, shorter transplantation vintage, and a more than two-drug immunosuppressive protocol. In subgroup analyses, the seroconversion rate was highest among KTR without MMF/MPS treatment (70%), treated with no more than two immunosuppressants (69.2%), treated without corticosteroid (66.7%), younger patients aged <54 years (63.2%), and those vaccinated with the mRNA-1273 vaccine (62.16%). The independent predictors of higher S-antibody titer among responders were younger age, treatment with no more than two immunosuppressants, and the mRNA-1273 vaccination. Conclusions: Our study confirmed a low rate of seroconversion after vaccination with the mRNA vaccine in KTR. The major modifiable determinants of humoral response were the composition of the immunosuppressive protocol, as well as the type of vaccine. The latter could be taken into consideration when initial vaccination as well as booster vaccination is considered in KTR.

## 1. Introduction

In the advent of coronavirus disease 2019 (COVID-19) the vaccination of kidney and other solid organ transplant recipients (SOTR) has emerged as a tool protecting this high-risk population, whose case fatality ratio for COVID-19 otherwise ranges between 13 to over 30% [1]. Despite some concerns related to the risk of inducing rejection, which can be triggered by the vaccine antigen or an associated adjuvant, or by more specific cellular and humoral cross reactivity between vaccine epitopes and allograft antigens, two mRNA vaccines (BNT162b2/Pfizer and mRNA-1273/Moderna) authorized by regulatory agencies are widely applied in SOTR [2,3]. These are non-adjuvanted vaccines where the mRNA (30 μg/dose in BNT162b2 and 100 μg/dose in mRNA-1273) is encapsulated in lipid nanoparticles; lipid nanoparticles possess natural adjuvant activity [4]. Clinical trials initiated after the outbreak of the COVID-19 epidemic proved the high efficacy of mRNA vaccines in the general population, reaching, after the second dose, 95% [2,3]. The results of our previous studies also showed a significant humoral post-vaccination response in the vast majority of dialyzed patients [5,6]. In contrast, kidney transplant recipients (KTR) demonstrated a markedly impaired seroconversion rate of 40–60% after two doses of mRNA vaccines [7,8,9]. Liver, heart, and lung transplant recipients exhibited a reduced response to mRNA-based vaccines as well [10,11,12]. In a recently published study by Stumpf et al., KTR not only demonstrated a low humoral response following two doses of mRNA vaccines, but also displayed substantial impairment of the cellular response [13]. Interestingly, some preliminary studies presented a better seroconversion rate in SOTR receiving mRNA-1273 [11,13,14]. These facts have mobilized the transplant community to evaluate the determinants of the response, and find measures to augment vaccine immunogenicity, given the likelihood that COVID-19 will remain a worldwide threat to the health of SOTR. Therefore, in our paper we aimed to evaluate the factors determining immunization in COVID-naïve KTR including different mRNA vaccines. We attempted to find any modifiable factors influencing the response.

## 2. Materials and Methods

### 2.1. Study Design

This longitudinal study was performed in a group of 243 KTR, and 50 patients without chronic kidney disease, who are managed by our institution and received vaccination with a two-dose mRNA vaccine: BNT162b2 (BionTech/Pfizer Comirnaty) or mRNA-1273 (Moderna), given according to the manufacturer’s recommendations. Neither any patient, nor the study team, had choice or influence regarding the type of mRNA vaccine, which was assigned according to the order of contacting the local vaccination point. KTR were considered eligible if they were at least 1 month after transplantation and had not been confirmed with SARS-CoV-2 infection in the past. Control patients were included if they had a confirmed estimated glomerular filtration rate–eGFR > 60 mL/min, had not been confirmed with SARS-CoV-2 infection in the past, and were vaccinated against COVID-19 with the same vaccines and schedule as KTR. Subjects were classified as having COVID-19 if (1) there was clear medical documentation with a positive SARS-CoV-2 PCR swab; (2) if seroconversion in nucleocapsid (N)- IgG specific antibodies was found in obligatory analyses performed before the first dose and the second dose of the vaccine. The main goal of the study was to analyze the seroconversion rate and titer magnitude of IgG antibodies directed against spike (S) SARS-CoV-2 antigen after the second dose of vaccination; and also to look for associations between them and potential predictors such as demographic, clinical and laboratory data, type of the vaccine, and immunosuppression regimen. Serum samples for anti-S and anti-N antibody titer were obtained before the first dose of the vaccine, and 14–21 days following the second one. Medical histories of study participants were extracted from their medical records. Transplantation vintage was defined as the time from the kidney transplantation until the baseline. The study was conducted according to the guidelines of the Declaration of Helsinki, and approved by the ethics committee of the Medical University of Gdansk (NKBBN/167/2021). The study is part of the ‘COVID-19 in Nephrology’ (COViNEPH) project focusing on the nephrological aspects of COVID-19, in particular epidemiology, prevention, disease course, and treatment registered on the ClinicalTrials.gov, identifier NCT04905862.

### 2.2. Anti-SARS-CoV-2 Antibodies Measurement

Quantitative measurement of specific IgG antibodies against trimeric S-protein was performed with a commercial chemiluminescent immunoassay (The LIAISON^®^ SARS-CoV-2 Trimetric-S IgG test, Diasorin, Italy) with a detection range of 1.85–800 AU/mL, as described previously [5]. Values over 800 were diluted to 1:20 to obtain an exact value. The assay presents a sensitivity of 98.7%, and specificity of 99.5%, and agreement with neutralization in microneutralization tests: PPA: 100%, NPA: 96.9%. Samples were interpreted as positive (seroconversion) or negative according to the manufacturer’s instructions, with a cutoff index value of >12 AU/mL. A conversion of AU/mL to binding antibody units (BAU/mL), that correlates with the WHO standard, is possible using the following equation: BAU/mL = 2.6*AU/mL. Nucleocapsid (N)-specific IgG antibodies were assessed using the commercially available Abbott Architect SARS-CoV-2 IgG 2-step chemiluminescent immunoassay. This assay presents a sensitivity/PPA of 100.0% and specificity/NPA of 99.63%. Samples were interpreted as positive (seroconversion) or negative with a cutoff index value s/c index of 1.4.

### 2.3. Statistical Analysis

Data were presented as a number (percentage) for categorical variables, and median (interquartile range; IQR) for continuous variables. A Chi-square test was used for categorical variables. Continuous variables were first tested for normal distribution using Shapiro-Wilk, and then compared by *t* test, if normally distributed, or by the Mann–Whitney test if abnormally distributed. Multivariable logistic stepwise regression was used to determine the independent factors associated with seroconversion in anti-s IgG antibodies, while multivariable linear regression was used to determine the independent factors associated with the titer of S-antibodies. Any variables that were at the significance level p less than 0.15 in univariable analyses were put in these models. All data was obtained using the software Statistica 13. *p* < 0.05 was considered significant.

## 3. Results

### 3.1. Patients Characteristic

Two hundred and forty-three KTR were screened; 43 individuals were excluded due to a history of COVID-19 and/or a positive test for anti-N antibody, and 2 KTR declined to participate; so, ultimately 198 subjects were eligible and included into the study. 56 patients withdrew, and thus 142 KTR were finally qualified to the per protocol analysis. The reasons for loss to follow-up, are described in Figure 1. The control group included 36 COVID-19–infection-naïve patients without chronic kidney disease. Patients’ demographics and clinical characteristics are detailed in Table 1.

### 3.2. Seroconversion in Anti-s IgG Antibodies

Of the 142 KTR who received both doses of either the mRNA-1273 or BNT162b2vaccine, 73 (51.41%) developed seroconversion in anti-s IgG antibodies as compared to 36 (100%) patients from the control group (*p* < 0.001). S-specific immune response in seroconverted KTR with a median (IQR) antibody IgG titer of 111 (33.90–327) AU/mL was lower than that observed in the seroconverted control patients of 815 (698.5–1440) (*p* < 0.001). In subgroup analyses, the seroconversion rate was highest among KTR without mycophenolate mofetil/Na (MMF/MPS) treatment (70%), treated with no more than two immunosuppressants (69.2%), treated without corticosteroid (66.7%), younger patients aged <54 years (63.2%), and vaccinated with mRNA-1273 vaccine (62.16%). Details are presented in Figure 2.

### 3.3. Determinants of The Seroconversion in Anti-s IgG Antibodies in KTR

The use of more than two immunosuppressive agents (*p* = 0.009), treatment with MMF/MPS (*p* = 0.02), shorter transplantation vintage (*p* = 0.002), and older age (*p* = 0.002) were predictors of no response to the vaccine in univariable analysis, and were therefore retained in the multivariable logistic regression model along with the variables for which the trend of association with the seroconversion was shown: CCI, induction treatment in history, serum creatinine level, and type of vaccine (Table 2).

### 3.4. Determinants of Anti-s IgG Titer Magnitude in KTR Responders

In univariable analysis, a statistically significant reduction in anti-s IgG titer in responders was associated with older age (Spearman R: −0.25; *p* = 0.029), use of MMF/MPS (*p* < 0.001), treatment with more than 2 immunosuppressants (*p* = 0.003), and vaccination with BNT162b2 mRNA vaccine (*p* = 0.043). In subgroup analysis, the highest titer magnitude was found among KTR without MMF/MPS treatment, vaccinated with mRNA-1273 vaccine, younger patients aged <54 years, and in those treated with no more than two immunosuppressants. Details are presented in Figure 3.

In the multivariable linear regression model, age, type of mRNA vaccine, and the amount of immunosuppressants maintained statistical significance (Table 4). This model was statistically significant (*p* < 0.001) and had a predictive capacity of 22.05% (adjusted R2) of the cases.

## 4. Discussion

We demonstrated that only 51.4% of COVID-19 naïve KTR achieved seroconversion. Participants were considered to have seroconverted if positive for IgG antibodies against trimeric spike-protein. In other words, a positive antibody test indicated that an immune response had occurred after vaccination. In addition, the magnitude of the response to vaccination was much lower as compared to the immunocompetent controls. The independent predictors of humoral response were the composition of the immunosuppressive protocol and the transplantation vintage. Significantly stronger immunization was noticed in patients receiving the mRNA-1273 vaccine as compared to BNT162b2. As in other studies, age was found to be an important factor in the humoral response, so the young people have an increased capacity to mount a humoral immune response compared to the older population [5,8,13]. We discuss below these results in relation to data from the literature.

In our study an immunosuppressive protocol consisting of more than two drugs was an independent predictor of a blunted humoral response to the vaccination. Additionally, we proved that short transplantation vintage was an independent risk factor of no response. It is quite convincing that these two factors may be coincident, since shortly after transplantation, patients usually receive a protocol consisting of three medications. On the other hand, patients with a long transplantation vintage receive not only lower doses of immunosuppressant, but quite frequently are maintained on two drugs. These two factors allow to some extent the reconstitution of immunological capacity. Our findings are in close agreement with two recent publications that demonstrated an impaired humoral and cellular immunity in KTR after vaccination, which correlated with the type and number of immunosuppressive agents [8,13]. Antimetabolites MMF/MPA seem to have a particularly unfavorable influence in this regard, which was also shown in a preliminary study by Boyarski [15]. This has also been observed in influenza vaccination [16]. In line with these observations, we found that the anti-s seroconversion rate after vaccination was highest among KTR without MMF/MPS treatment, reaching 70%. Multivariable analyses indicated that these agents affect not only anti-S serostatus, but also the magnitude of the immune response in the responders. Despite these findings, minimizing or withholding the antiproliferative agent at the time of a vaccine booster in patients with a failed vaccine antibody response is not recommended in KTR; although in long-standing liver transplant recipients it may be a reasonable approach after an individualized discussion of the risks and benefits [17,18].

Unlike other studies, we have found no clear association between steroid use or immunosuppression induction and immunogenicity of the vaccines, although in strata analysis the seroconversion rate in patients without steroids was 66.7% [8,13]. Despite the slight inhibitory effect of anti-thymocyte globulin or anti-IL-2 receptor antibodies on the rate of anti-S seroconversion, the difference did not reach statistical significance. Perhaps this was due to the large differences in strata sample size, related to the infrequent use of the induction, and the small proportion of our patients not receiving steroids.

The question is whether the low humoral response in KTR is related only to immunosuppression. To some extent it can be explained as a consequence of immunosuppressive therapy, but other causes such as the type of vaccine should be considered. We confirmed the results published quite recently in *Lancet Reg Health Eur* demonstrating that the BNT162b2 vaccine may be an independent risk factor for worse humoral immunity in KTR [13]. Stumpf et al. noticed a blunted response to this type of mRNA vaccine also in hemodialyzed patients, but interestingly, not in the control group consisting of immunocompetent subjects. Similar differences in the vaccine responses of KTR were reported by Boyarski after the first dose, but less pronounced after a boost vaccination [14,15].

The data presented by Firket et al., and confirmed recently by our team (unpublished data), shows that natural exposure to the virus seems to be a stronger stimulus than vaccination in terms of the formation of immunological memory, and the production of antibodies upon repeated contact with the antigen (reaction to the vaccination) [19]. The difference in the immune response, as compared to natural infection, may be explained by the antigenic stimuli provided by the whole SARS-CoV-2- in comparison to the spike-protein of only vaccine. In line with these observations, we found here that the type and amount of antigenic vaccine stimulation played an important role for KTR. Seroconversion rates and antibody titer magnitude were significantly higher after mRNA-1273 as compared to BNT162b2. One of the explanations of the higher mRNA-1273 vaccine immunogenicity could be the three-times-higher dose of mRNA, and there may be some other vaccine-related factors such as its stability. The demonstration that the type of vaccine is a modifiable factor of the response to vaccination is of importance when planning vaccination particularly in patients with a short transplantation vintage, receiving more than a two-drug immunosuppression, and elder people. There are many types of vaccines against COVID-19 that induce an immune response by different mechanisms, and create in this way hope for their higher efficacy in immunocompromised individuals. These include viral vector-based vaccines (e.g., AZD122 (AstraZeneca), JNJ78436735/Ad26.COV2.S (Janssen), and Sputnik V (Gamaleya) or the vaccines that are in the last phase of clinical trials, e.g., adjuvant protein subunit vaccines containing recombinant spike protein, NVX-CoV2373 (Novavax) and vaccines containing whole inactivated virus, VLA2001 (Valneva) [4,20]. The last type of the above vaccines is especially hopeful. Because the recombinant protein vaccines use only a protein fragment of receptor binding domain as the antigen, they may have lower immunogenicity than the whole-pathogen vaccine candidates containing more than 20 immunoreactive epitopes [21,22]. In a randomized phase 1/2 clinical trial with Valneva, antigen-specific interferon-γ T-cells reactive were observed against the spike, membrane, and nucleocapsid proteins [21]. Of course, take into account the fact that not all viral epitopes of the virus need to be immunogenic [23]. Research in this area is currently underway. For example, the immunogenicity of 5 virus epitopes from membrane glycol-protein (MGP) and non-structure protein-13 (NSP13) was validated on the basis of their ability to elicit peptide-specific T cells capable of recognizing and killing SARS-CoV2-expressing target cells [24]. Another study has identified ORF9b, N and M.ext/M proteins epitopes as promising candidates for a multi-epitope vaccine design [25]. The safety and clinical effectiveness of multi-epitope vaccines requires further studies.

The above considerations may contribute to finding schemes enhancing immune responses in immunocompromised patients, and securing them against the current and future circulating variants of SARS-CoV-2. Booster third vaccinations may be required, either to stimulate waning immunity, or to expand the breadth of immunity to SARS-CoV-2 variants [26]. KTR whose immunological response to two doses of mRNA vaccines was limited are considered not only for a third dose of homologous vaccine, but also for primary heterologous vaccine schedules or a heterologous booster. Studies exploring alternative sequence booster strategies (e.g., viral vector after nucleic acid platform), additional vaccine doses after a complete SARS-CoV-2 vaccine series (wild-type boost), or variant booster doses after wild-type primary vaccination are ongoing. Additionally, the interval between prime and boost probably has a critical role. In SOTR the combination of two vaccine strategies that offer complementary stimulation of different immune pathways may more effectively induce long-lasting B cell responses and potent T cell responses [27]. The heterologous vaccine schedules, however, might have some short-term disadvantages inducing greater systemic reactogenicity following the boost dose than their homologous counterparts, as shown by preliminary studies [28,29].

Our study is one of the first to analyze factors which may influence the humoral response to vaccination in KTR. Our sample seems to be “representative” of the KTR from Poland. Patients from many regions of the country are under the outpatient control of our institution. We use standard immunosuppression protocols, in line with the recommendations of the Polish Transplantation Society. Patients were randomly enrolled in the study in the order of their reporting to the vaccination point. The strength of our study is also the use of a highly sensitive test using the S-trimer antigen demonstrating almost 100% compliance with neutralization tests, and the exclusion of patients who have been infected with SARS-CoV-2. Thanks to the determination of anti-N antibodies before the first and second vaccination, we excluded all subjects who could have asymptomatically suffered from COVID-19, which has a significant impact on the immunogenicity of vaccines [19,30].

## 5. Limitations

One limitation is that we only tested humoral responses. The cellular part of the adaptive immune system plays a role in protection from COVID-19 which is not reflected in our investigation. To what extent cellular immunity, in the absence of detectable antibodies, is able to prevent severe infection in SOTR is yet to be determined [22]. Secondly, limitations of the study also include the observational, non-randomized study character, and the selection bias towards patients and controls interested in SARS-CoV-2 vaccination. Thirdly, there were no people vaccinated with mRNA-1273 vaccine in the control group, which makes it impossible to compare our conclusions regarding this vaccine in immunocompetent subjects. It should also be remembered that only two mRNA vaccines were tested in our study, so any comparisons with other types of vaccines raised in the discussion are only speculative and are an incentive for further research. Finally, we did not analyze the effect on seroconversion of the primary nephropathies that led to kidney insufficiency in our patients. Unfortunately, most of them were unknown. There were probably patients with autoimmune, immunodeficiency diseases, and it cannot be ruled out that some of these affect the humoral response.

## 6. Conclusions

In conclusion, we would like to underline our clinically relevant observations: (1) KTR demonstrate an impaired humoral immunity after vaccination against COVID-19, (2) the independent predictors of no response were elder age, shorter transplantation vintage, and a more than two-drug immunosuppressive protocol, (3) the independent predictors of antibody titer among responders were age, number of drugs in the immunosuppressive protocol, and type of mRNA vaccines, among which mRNA-1273 seems to show greater immunogenicity.

## Figures and Tables

**Figure 1 vaccines-09-01165-f001:**
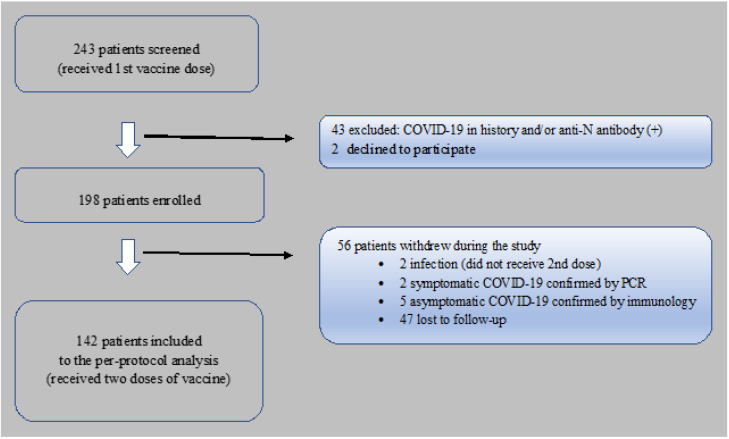
Flow chart of kidney transplant recipients vaccinated against COVID-19, screened and included to the per-protocol analysis.

**Figure 2 vaccines-09-01165-f002:**
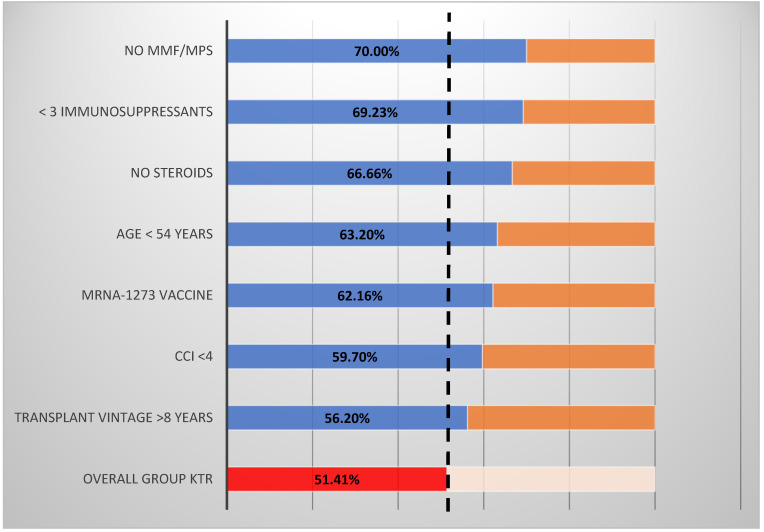
Strata analyses of anti-s IgG seroconversion rate in the study group. Legend: MMF/MPS, mycophenolate mofetil/Na; CCI; Charlson comorbidity index; KTR, kidney transplant recipients. Seroconversion rate (blue/red columns). Seroconversion in complimentary strata were as follows: MMF/MPS + (46.4%); <3 immunosuppressants (44.6%); steroids + (50%); age > 54 (39.1%); BNT162b2 + (47.6%); CCI > 4 (49.1%); transplantation vintage < 8 years (39.1%).

**Figure 3 vaccines-09-01165-f003:**
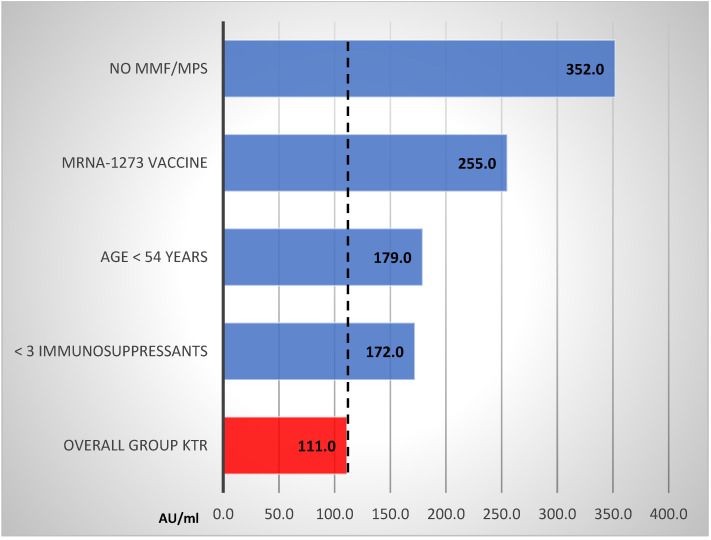
Strata analyses of anti-s IgG titer (AU/mL) in KTR responders. Legend: MMF/MPS; mycophenolate mofetil/Na.

**Table 1 vaccines-09-01165-t001:** Demographic and clinical characteristic study and control group.

	Study Group *n* = 142	Control Group *n* = 36	*p*-Value
Age years median (IQR)	54 (43–63)	48 (45–62)	ns
Sex male *n* (%)	83 (58.45)	21 (58.3)	ns
CCI median (IQR)	4 (2–5)	0.5 (0–1)	<0.001
Serum creatinine mg/dl median (IQR)	1.35 (1.12–1.7)		
BMI kg/m^2^ median (IQR)	25 (22.55–28.37)		
Primary nephropathy *n* (%) Unknown Glomerulonephritis ADPKD Other	36 (25.35) 36 (25.35) 21 (14.79) 35 (24.65)		
Transplant vintage years median (IQR)	8 (3.5–15)		
Deceased donor *n* (%)	133 (93.7)		
Immunosuppression protocol *n* (%) Protocol without steroids Protocol without MMF/MPS Protocol with induction	12 (8.5) 30 (21.13) 37 (26.06)		
mRNA-1273 vaccination *n* (%)	37 (26.06)	0 (0)	0.002
mRNA BNT162b2 vaccination *n* (%)	105 (73.94)	36 (100)	0.002

Legend: CCI; Charlson comorbidity index; BMI, body mass index; ADPKD, autosomal dominant polycystic kidney disease; MMF/MPS, mycophenolate mofetil/Na.

**Table 2 vaccines-09-01165-t002:** Univariable analysis of predictors for anti-s IgG seroconversion.

	Responders *n* = 73	Nonresponders *n* = 69	*p*-Value
Age years median (IQR)	48.0 (40–61)	58.0 (50–66)	0.002
Sex male *n* (%)	44 (60.3)	39 (56.5)	0.65
BMI kg/m^2^ median (IQR)	25.01 (23.11–28.37)	25.35 (22.49–28.41)	0.33
CCI median (IQR)	3 (2–5)	4 (3–6)	0.079
Diabetes *n* (%)	16 (21.92)	20 (28.98)	0.33
Transplant vintage years median (IQR)	10.0 (6–19)	7.0 (2.5–12)	0.002
Deceased donor *n* (%)	69 (94.52)	64 (92.75)	0.67
Serum creatinine mg/dl median (IQR)	1.31 (1.03–1.58)	1.39 (1.14–1.77)	0.13
Induction in history *n* (%)	16 (21.9)	23 (33.3)	0.13
>2 drugs immunosuppression *n* (%)	46 (65.83)	57 (82.61)	0.009
Corticosteroids *n* (%)	65 (89.04)	65 (94.2)	0.27
MMF/MPS *n* (%)	52 (71.2)	60 (86.9)	0.02
mRNA-1273 vaccine *n* (%)	23 (31.5)	14 (20.3)	0.13

*Legend: CCI; Charlson comorbidity index; BMI, body mass index; MMF/MPS, mycophenolate mofetil/Na.* The use of more than two immunosuppressive agents (*p* = 0.01), shorter transplantation vintage (*p* = 0.003), and older age (*p* < 0.001) maintained statistical significance in the multivariable analysis (Table 3). This regression analysis was statistically significant (χ2 = 29.95, *p* < 0.001). The Hosmer–Lemeshow test revealed a good fit (χ2 = 7.09), and the Nagelkerke R2 effect size (R2 = 0.25) demonstrated good predictive efficacy.

**Table 3 vaccines-09-01165-t003:** Multivariable logistic regression analysis of factors affecting anti-s seroconversion.

	Coefficient	*p-*Value	OR (95% CI)
Age	−0.059	<0.001	0.94 (0.91–0.97)
Transplant vintage	0.081	<0.003	1.08 (1.03–1.14)
>2 drug immunosuppression	−1.136	<0.01	0.32 (0.13–0.77)

Legend: OR, odds ratio; CI, confidence interval.

**Table 4 vaccines-09-01165-t004:** Multivariable linear regression analysis of factors involved in determining anti-S IgG antibody titer.

	Coefficient	Standard Error	*p*-Value
Age	−5.42	1.86	0.005
>2 immunosuppressants	−193.38	47.19	0.001
mRNA-1273 vaccine	91.55	47.33	0.05

## Data Availability

Detailed data are available upon request from corresponding author.

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
