# Peer review of "Predictors of Humoral Response to mRNA COVID19 Vaccines in Kidney Transplant Recipients: A Longitudinal Study—The COViNEPH Project"

_vaccines, 2021, doi:10.3390/vaccines9101165_

Round 1

Reviewer 1 Report

The manuscript of Alicja Dębska-Ślizień et. al. presents an important contribution to the understanding of mRNA based anti-COVID19 vaccines induced immunogenicity in the immunocompromised human patients, recipients of kidney transplant (KTR).  The research presented in this manuscript is of high importance for further evaluating the major factors controlling the seroconversion in anti-trimeric S (spike protein) IgG antibodies (the main component of the anti-COVID 19 vaccines). 

In summary the main contribution of the presented research is the discovery of  a low immunological response in the KTR patients; specifically, only 51.4% of COVID-19 naïve KTR achieved seroconversion in anti-S IgG antibodies. Importantly, the authors show experimental data that supporting the fact that the magnitude of the response to vaccination was statistically significant lower as compared to the immunocompetent controls.

The most important contribution of the presented research is the analysis conducted to highlight the independent predictors of humoral response, mainly the composition of the immunosuppressive protocol, and the transplantation vintage. Remarkably, the results presented show a significantly stronger immunization in patients  receiving the mRNA-1273 vaccine as compared to BNT162b2.  Moreover, the authors produced complementary results with other published studies, showing that age is an important factor in the humoral response. As such, the data presented herein show that the young people have an increased capacity to mount a humoral immune response compared to the older population.

Remarkably, the authors are concluding the paper with additional analyses on the limitations of their own study. One important mentioned limitation is the one-sided analysis of the humoral immune response, omitting the cell-based, T cell mediated adaptive immune response.  Especially, the authors emphasize to investigate  how the cellular mediated immune response could limit the severe infection in SOTR.

Other mentioned limitations are related to the study itself including the observational, non-randomized study character and the selection bias towards patients and controls interested in SARS-CoV-2 vaccination. 

I recommend the paper to be published in the "Vaccines" journal in the presented format.  

Author Response

RE: Thank you for the positive evaluation of our study.

Reviewer 2 Report

The paper investigates the features of the humoral immune response of patients with kidney transplants undergoing various courses of immunosuppressive therapy and at different times after kidney transplantation.

The manuscript presents interesting results on the effect of immunosuppressants on seroconversion against two recently released COVID19 RNA vaccines. The authors found a suppressive effect on seroconversion of mycophenolate mofetil/Na (MMF/MPS), treatment with corticosteroids, by using more than two immunosuppressants (p = 0.009), the effect of the patient's age (p = 0.002), and the type of vaccine. A particularly important finding is that patients vaccinated with the mRNA-1273 vaccine (Moderna) develop seroconversion more often than BNT162b2 (Pfizer) vaccine. In the discussion, the authors recommend using vaccines that give a greater variety of antibodies to different epitopes and suggest that the use of vaccines with multiple protein epitopes (inactivated viruses or viral proteins) will increase seroconversion. However, at the same time, it is not discussed that this can also increase the risk of complications and, perhaps, could switch the immune response to the production of IgG antibodies to immunogenic but insignificant epitopes that do not have a high protective effect. In addition, future readers, perhaps, would like the authors to at least, briefly explain the concept of seroconversion, and why authors are interested in seroconversion.

The work is undoubtedly interesting, significant, and of great social and scientific importance. However, I would like to give some suggestions for improving the manuscript.

Major comments

In the discussion section, the authors described the limitations of their study. However, it would be wonderful to put the “study limitations” in a separate subsection and discuss additional issues. For example, only two vaccines were included in the studies, but the authors draw conclusions and comparisons with other vaccines without citing specific results. The authors do not mention the diseases that led to kidney transplantation. It remains unclear how humoral immunity and seroconversion are affected by these serious diseases from which patients continue to suffer despite kidney transplantation. Indeed, there were probably patients with autoimmune, immunodeficiency diseases and suffering from severe forms of diabetes, etc., which affects the effectiveness of the humoral response and seroconversion. In addition, there were no volunteers vaccinated with mRNA-1273 in the control group, how does this affect the author's conclusions?

In the data and tables, it is difficult to find data for the control group, as well as to understand what the results were in comparison with the control group.

The conclusions are vague and too general. Conclusions should be made formally and concretely, however, the authors limited themselves to general phrases. The feeling that the authors are recommending that we reread the article again instead of clearly summarizing the conclusions from their data at the end of the manuscript in a concise manner.

Minor comments

Line 24: Please explain the sentence and what it means “modifiable predictors of humoral response…” Are you sure you really need this sentence in your abstract? Your abstract has already exceeded the allowed word count. It should be drastically shortened to the required number of words in the Vaccines MDPI Journal.

Line 28: please check it “in 142 infection naïve KTR … -> “in 142 infected naïve KTR”

Line 51: “concerns related with the risk” -> concerns related to the risk

Line 84: The abbreviation must be disclosed – “eGFR”

Lines 118, 126: “Data was…” please check it (see reference below)

https://www.onlinegrammar.com.au/top-10-grammar-myths-data-is-plural-so-must-take-a-plural-verb/

Please in the chapter "Materials and Methods" explain the term " the transplantation vintage"

Lines 280-282: Please, correct the grammar in the sentence, clarify the meaning and simplify this sentence. “KTR whose immunological response to two doses of mRNA vaccines was limited are considered not only for a third dose of homologous vaccine, but also for primary heterologous vaccine schedules or a heterologous booster”.

Author Response

Reviewer: 2

a.    In the discussion, the authors recommend using vaccines that give a greater variety of antibodies to different epitopes and suggest that the use of vaccines with multiple protein epitopes (inactivated viruses or viral proteins) will increase seroconversion. However, at the same time, it is not discussed that this can also increase the risk of complications and, perhaps, could switch the immune response to the production of IgG antibodies to immunogenic but insignificant epitopes that do not have a high protective effect. 

RE: As suggested by the reviewer, a relevant comment was added to the discussion as below:

In a randomized phase 1 /2 clinical trial with Valneva, antigen-specific interferon-γ T-cells reactive were observed against the spike, membrane and nucleocapsid proteins [21]. Of course, take into account the fact that not all viral epitopes of the virus need to be immunogenic [23]. Research in this area is currently underway. For example, the immunogenicity of 5 virus epitopes from membrane glycol-protein (MGP) and non-structure protein-13 (NSP13) was validated on the basis of their ability to elicit peptide-specific T cells capable of recognizing and killing SARS-CoV2-expressing target cells [24]. The another has identified ORF9b, N and M.ext/M proteins epitopes as promising candidates for the multi-epitope vaccine design [25]. The safety and clinical effectiveness of multi-epitope vaccines require further studies.

b.    In addition, future readers, perhaps, would like the authors to at least, briefly explain the concept of seroconversion, and why authors are interested in seroconversion.
RE: As suggested by the reviewer, a relevant comment was added to the discussion as below:
“Participants were considered to have seroconverted if positive for IgG antibodies against trimeric spike-protein. In other words a positive antibody test indicates an immune response has occurred after vaccination”.

c.    It would be wonderful to put the “study limitations” in a separate subsection and discuss additional issues. 
•    Only two vaccines were included in the studies, but the authors draw conclusions and comparisons with other vaccines without citing specific results. 
•    The authors do not mention the diseases that led to kidney transplantation. It remains unclear how humoral immunity and seroconversion are affected by these serious diseases from which patients continue to suffer despite kidney transplantation. Indeed, there were probably patients with autoimmune, immunodeficiency diseases and suffering from severe forms of diabetes, etc., which affects the effectiveness of the humoral response and seroconversion. 
•    In addition, there were no volunteers vaccinated with mRNA-1273 in the control group, how does this affect the author's conclusions?

RE: The “study limitations” was put in a separate subsection. Additional considerations have been included in the Limitations section as directed.

“Thirdly, there were no people vaccinated with mRNA-1273 vaccine in the control group, which makes it impossible to compare our conclusions regarding this vaccine in im-munocompetent subjects. It should also be remembered that only two mRNA vaccines were tested in our study, so any comparisons with other types of vaccines raised in the discussion are only speculative and are an incentive for further research. Finally, we did not analyze the effect on seroconversion of the primary nephropathies that led to kidney insufficiency in our patients. Unfortunately, most of them were unknown. There were probably patients with autoimmune, immunodeficiency diseases and it cannot be ruled out that some of these affect the humoral response”.

d.    In the data and tables, it is difficult to find data for the control group, as well as to understand what the results were in comparison with the control group.

RE: The characteristics of the control group and the comparison to the study group are presented in Table 1. Data on the seroconversion rate in control group and comparison to the study group are presented in results section 3.2. Due to the 100% seroconversion rate in the control group, the multivariate analysis of seroconversion predictors was performed only in the study group (Figure 2, Table 2 and 3). The comparison of the anti-s antibody titer magnitude among the responders from study and control groups is presented in results section 3.2. 

e.    The conclusions are vague and too general. Conclusions should be made formally and concretely, however, the authors limited themselves to general phrases. 

RE: It was done as requested. The current Conclusions section is as follows:         

“In conclusion, we would like to underline our clinically relevant observations: 1) KTR demonstrate an impaired humoral immunity after vaccination against COVID-19, 2) the independent predictors of no response were: elder age, shorter transplantation vintage, a more than two-drug immunosuppressive protocol, 3) the independent predictors of an-ti-body titer among responders were: age, number of drugs in the immunosuppressive protocol, and type of mRNA vaccines among which mRNA-1273 seems to show greater immunogenicity”.

f.    Your abstract has already exceeded the allowed word count. It should be drastically shortened to the required number of words in the Vaccines MDPI Journal
RE: The manuscript abstract has been shortened significantly.
---------------------------------------------------------------------------------

g.    Line 28: please check it “in 142 infection naïve KTR … -> “in 142 infected naïve KTR”
h.    Line 51: “concerns related with the risk” -> concerns related to the risk
i.    Line 84:The abbreviation must be disclosed – “eGFR”
j.    Lines 118, 126: “Data was…” please check it (see reference below) https://www.onlinegrammar.com.au/top-10-grammar-myths-data-is-plural-so-must-take-a-plural-verb/

RE: The comments from the points g-j were answered by a native speaker who improved the work (below).

k.    please in the chapter "Materials and Methods" explain the term " the transplantation vintage"
RE: It was done as requested.

l.    Lines 280-282: Please, correct the grammar in the sentence, clarify the meaning and simplify this sentence. “KTR whose immunological response to two doses of mRNA vaccines was limited are considered not only for a third dose of homologous vaccine, but also for primary heterologous vaccine schedules or a heterologous booster”.

RE: The comments from this point were answered by a native speaker who improved the work (below).
---------------------------------------------------------------------------------

Answers of Native Speaker Adam Green: 

Dear Sir/Madam,
My name is Adam Green and I work here in Poland as an English teacher and proof-reader; I would like to state that I have looked at, and corrected, the paper including the proposed changes which you as  the Editor asked for. Additionally, my comments  on  the Reviewers point by point are placed below.  
Faithfully, Adam Green.

g. Line 28: please check it “in 142 infection naïve KTR … -> “in 142 infected naïve KTR”
Re: surely “infection” is a better choice here than “infected” ?

h. Line 51: “concerns related with the risk” -> concerns related to the risk
Re: these are both reasonable. 

i.    Line 84:The abbreviation must be disclosed – “eGFR”  
Re: it is done in the main text

j.  Lines 118, 126: “Data was…” please check it (see reference below) https://www.onlinegrammar.com.au/top-10-grammar-myths-data-is-plural-so-must-take-a-plural-verb/ 
Re: this is a well-known example, and all around the world people/scientists/researchers are conventionally using “data was”. The Reviewer is correct, as we all know, that “data” is plural (as compared to “datum”, singular) and so “data were” would also be fine. At the moment, the widely accepted majority view is that “data was” rules.
It is a paradox, but surely the Reviewer is aware of this exception? 
(As an exercise in pedantry, I believe that the Reviewer would enjoy “The Army is required to . . . “ or perhaps would prefer “The Army are required to . . . “ ) 

l. Lines 280-282: Please, correct the grammar in the sentence, clarify the meaning and simplify this sentence. “KTR whose immunological response to two doses of mRNA vaccines was limited are considered not only for a third dose of homologous vaccine, but also for primary heterologous vaccine schedules or a heterologous booster”. 
Re: I  suggest  leaving the sentence as it is. This is fine grammatically, and the sense is clear.

Reviewer 3 Report

Estimated Authors,

first at all, thank you for the opportunity to review this interesting article from the study group lead by Alicja Dębska-Ślizień, and reporting on the immunogenicity of SARS-CoV-2 vaccines based on mRNA in Kidney Transplantation Recipients.

The authors have identified an extensive lack of efficacy of vaccines among these individuals, and have also characterized a series of potential risk factors for vaccination failure (at least, in terms of IgG induction). Interestingly, younger age, being "naïve" in terms of immuno-suppresant therapy, and having received a mRNA-1273 vaccine rather than Pfizer one were associated with a better outcome.

In fact, these results are both interesting and significant. However, I've a series of doubts about the present stage of this paper, and more precisely:

1) the overall quality of the English text (that is largely correct in terms of grammar) may benefit from some further editing. For example: "

Row48-49 … “In the advent of coronavirus disease 2019 (COVID-19) the vaccination of kidney and other solid organ transplant recipients (SOTR) has emerged as a tool protecting this high-risk population with COVID-19 mortality varying between 13 to over 30% [1]” --> “In the advent of coronavirus disease 2019 (COVID-19) the vaccination of kidney and other solid organ transplant recipients (SOTR) has emerged as a tool protecting this high-risk population, whose case fatality ratio for COVID-19 otherwise ranges between 13 to over 30% [1]"; similarly the following sentence: "Despite some concerns related with the risk of inducing rejection that can be triggered by the vaccine antigen or an associated adjuvant, or by more specific cellular and humoral cross reactivity between vaccine epitopes and allograft antigens". 

2) How the sample was ultimately collected remains unclear. More precisely: we have 243 potential subjects, that were initially reduced to 198, and eventually to 142. Even though the information about the selection procedure was provided in Figure 1, some glimpses must be included in the main text. Without such information, Readers may fail to correctly understand how the study population was ultimately reduced to the final sample.

3) strictly associated with point 2), I would recommend Authors to discuss whether the sample may be considered "representative" of the KTR from Poland. This is particularly significant in order to reaffirm (or discuss) whether the multivariable analysis may be retained as reliable or not. In my opinion, no significant reworking is necessary; Authors should only discuss such issue in the final section of the paper.

4) Even though results are properly reported, I've noticed some inconsistencies in the data reporting in terms of labeling of Tables and Figures. For example: 

Please focus on the last row of table 2; in univariable analysis, mRNA-1273 vaccine was employed among 31.5% of responders, and only among 20.3% of non responders, with a p value 0.13. Reasonably, the factor was included in the multivariable model. However, in Table 4 the label reports “BNT162b2 mRNA vaccine”, while even figure 3 reports the label “mRNA 1273 vaccine”. In other words, it seems a little bit misleading. Please modify captions for Figure 3 and Table 4 in order to make them consistent.

Author Response

Reviewer 3

a. The overall quality of the English text (that is largely correct in terms of grammar) may benefit from some further editing. For example: "
•    Row48-49 … “In the advent of coronavirus disease 2019 (COVID-19) the vaccination of kidney and other solid organ transplant recipients (SOTR) has emerged as a tool protecting this high-risk population with COVID-19 mortality varying between 13 to over 30% [1]” --> “In the advent of coronavirus disease 2019 (COVID-19) the vaccination of kidney and other solid organ transplant recipients (SOTR) has emerged as a tool protecting this high-risk population, whose case fatality ratio for COVID-19 otherwise ranges between 13 to over 30% [1]"; 
•    similarly the following sentence: "Despite some concerns related with the risk of inducing rejection that can be triggered by the vaccine antigen or an associated adjuvant, or by more specific cellular and humoral cross reactivity between vaccine epitopes and allograft antigens". 

RE: These comments were answered at the end by a native speaker who corrected the work

b. How the sample was ultimately collected remains unclear. More precisely: we have 243 potential subjects, that were initially reduced to 198, and eventually to 142. Even though the information about the selection procedure was provided in Figure 1, some glimpses must be included in the main text. Without such information, Readers may fail to correctly understand how the study population was ultimately reduced to the final sample.

RE: The recruitment description used has been added as below: 

Two hundred and forty three KTR were screened; 43 individuals were excluded due to history of COVID-19 and/or positive test for anti-N antibody, 2 KTR declined to par-ticipate so ultimately 198 subjects were eligible and included into the study. 56 patients withdrew and 142 KTR were finally qualified to the per protocol analysis. The reasons for loss to follow-up, are described in Figure 1.

c. I would recommend Authors to discuss whether the sample may be considered "representative" of the KTR from Poland. This is particularly significant in order to reaffirm (or discuss) whether the multivariable analysis may be retained as reliable or not. In my opinion, no significant reworking is necessary; Authors should only discuss such issue in the final section of the paper.

RE: In the final section of Discussion the relevant comment was added as follows:

“Our sample seems to be considered "representative" of the KTR from Poland. Patients from all regions of the country are under outpatient control of our institution. It uses standard immunosuppression protocols, in line with the recommendations of the Polish Transplantation Society. Patients were randomly enrolled in the study in the order of their reporting to the vaccination point”.

d. Even though results are properly reported, I've noticed some inconsistencies in the data reporting in terms of labeling of Tables and Figures. For example:  please focus on the last row of table 2; in univariable analysis, mRNA-1273 vaccine was employed among 31.5% of responders, and only among 20.3% of non-responders, with a p value 0.13. Reasonably, the factor was included in the multivariable model. However, in Table 4 the label reports “BNT162b2 mRNA vaccine”, while even figure 3 reports the label “mRNA 1273 vaccine”. In other words, it seems a little bit misleading. Please modify captions for Figure 3 and Table 4 in order to make them consistent.

RE: It was done as requested.

-------------------------------------------------------------------------------
Answers of Native Speaker Adam Green: 

Dear Sir/Madam,
My name is Adam Green and I work here in Poland as an English teacher and proof-reader; I would like to state that I have looked at, and corrected, the paper including the proposed changes which you as  the Editor asked for. Additionally, my comments  on  the Reviewers point by point are placed below.  
Faithfully, Adam Green.

a. The overall quality of the English text (that is largely correct in terms of grammar) may benefit from some further editing. For example: "
●    Row48-49 … 
“In the advent of coronavirus disease 2019 (COVID-19) the vaccination of kidney and other solid organ transplant recipients (SOTR) has emerged as a tool protecting this high-risk population with COVID-19 mortality varying between 13 to over 30% [1]” --> 

“In the advent of coronavirus disease 2019 (COVID-19) the vaccination of kidney and other solid organ transplant recipients (SOTR) has emerged as a tool protecting this high-risk population, whose case fatality ratio for COVID-19 otherwise ranges between 13 to over 30% [1]";  

Re: - Suggested by the Reviever 2  part of the sentence “whose case fatality ratio for COVID-19 otherwise ranges” replaced the  previous part of the original sentence “this high-risk population with COVID-19 mortality varying”.
Both of these options are good, there was nothing wrong with the original version. However, we replaced the part of the sentence to the suggested version.

●    Similarly, the following sentence: "Despite some concerns related with the risk of inducing rejection that can be triggered by the vaccine antigen or an associated adjuvant, or by more specific cellular and humoral cross reactivity between vaccine epitopes and allograft antigens".  

Re: The sentence was longer, not as the Reviewer  2 cited above,  and proper as far as the grammar, and especially sense, is concerned. 
Despite some concerns related with the risk of inducing rejection that can be triggered by the vaccine antigen or an associated adjuvant, or by more specific cellular and humoral cross reactivity between vaccine epitopes and allograft antigens two mRNA vaccines (BNT162b2/Pfizer;  and mRNA-1273/Moderna;) authorized by regulatory agencies are widely applied in SOTR [2,3]